# AdaGCN: Adaboosting Graph Convolutional Networks into Deep Models

**Ke Sun**
Zhejiang Lab
Key Lab. of Machine Perception (MoE), School of EECS, Peking University
ajksunke@pku.edu.cn

**Zhanxing Zhu**[*]
Beijing Institute of Big Data Research, Beijing, China
zhanxing.zhu@pku.edu.cn

**Zhouchen Lin**[*]
Key Lab. of Machine Perception (MoE), School of EECS, Peking University
Pazhou Lab, Guangzhou, China
zlin@pku.edu.cn

## ABSTRACT

The design of deep graph models still remains to be investigated and the crucial part is how to explore and exploit the knowledge from different hops of neighbors in an efficient way. In this paper, we propose a novel RNN-like deep graph neural network architecture by incorporating AdaBoost into the computation of network; and the proposed graph convolutional network called AdaGCN (Adaboosting Graph Convolutional Network) has the ability to efficiently extract knowledge from high-order neighbors of current nodes and then integrates knowledge from different hops of neighbors into the network in an Adaboost way. Different from other graph neural networks that directly stack many graph convolution layers, AdaGCN shares the same base neural network architecture among all "layers" and is recursively optimized, which is similar to an RNN. Besides, We also theoretically established the connection between AdaGCN and existing graph convolutional methods, presenting the benefits of our proposal. Finally, extensive experiments demonstrate the consistent state-of-the-art prediction performance on graphs across different label rates and the computational advantage of our approach AdaGCN [1].

## 1 INTRODUCTION

Recently, research related to learning on graph structural data has gained considerable attention in machine learning community. Graph neural networks (Gori et al., 2005; Hamilton et al., 2017; Veličković et al., 2018), particularly graph convolutional networks (Kipf & Welling, 2017; Defferrard et al., 2016; Bruna et al., 2014) have demonstrated their remarkable ability on node classification (Kipf & Welling, 2017), link prediction (Zhu et al., 2016) and clustering tasks (Fortunato, 2010). Despite their enormous success, almost all of these models have *shallow model architectures* with only two or three layers. The shallow design of GCN appears counterintuitive as deep versions of these models, in principle, have access to more information, but perform worse. Oversmoothing (Li et al., 2018) has been proposed to explain why deep GCN fails, showing that by repeatedly applying Laplacian smoothing, GCN may mix the node features from different clusters and makes them indistinguishable. This also indicates that by stacking too many graph convolutional layers, the embedding of each node in GCN is inclined to converge to certain value (Li et al., 2018), making it harder for classification. These shallow model architectures restricted by oversmoothing issue

---

[*]Corresponding author.
[1]Code is available at https://github.com/datake/AdaGCN.

limit their ability to extract the knowledge from high-order neighbors, *i.e., features from remote hops of neighbors for current nodes*. Therefore, it is crucial to design deep graph models such that high-order information can be aggregated in an effective way for better predictions.

There are some works (Xu et al., 2018b; Liao et al., 2019; Klicpera et al., 2018; Li et al., 2019; Liu et al., 2020) that tried to address this issue partially, and the discussion can refer to Appendix A.1. By contrast, we argue that a key direction of constructing deep graph models lies in the efficient exploration and effective combination of information from different orders of neighbors. Due to the apparent *sequential relationship* between different orders of neighbors, it is a natural choice to incorporate boosting algorithm into the design of deep graph models. As an important realization of boosting theory, AdaBoost (Freund et al., 1999) is extremely easy to implement and keeps competitive in terms of both practical performance and computational cost (Hastie et al., 2009). Moreover, boosting theory has been used to analyze the success of ResNets in computer vision (Huang et al., 2018) and AdaGAN (Tolstikhin et al., 2017) has already successfully incorporated boosting algorithm into the training of GAN (Goodfellow et al., 2014).

In this work, we focus on incorporating AdaBoost into the design of deep graph convolutional networks in a non-trivial way. Firstly, in pursuit of the introduction of AdaBoost framework, we refine the type of graph convolutions and thus obtain a novel RNN-like GCN architecture called AdaGCN. Our approach can efficiently extract knowledge from different orders of neighbors and then combine these information in an AdaBoost manner *with iterative updating of the node weights*. Also, we compare our AdaGCN with existing methods from the perspective of both architectural difference and feature representation power to show the benefits of our method. Finally, we conduct extensive experiments to demonstrate the consistent state-of-the-art performance of our approach across different label rates and computational advantage over other alternatives.

## 2 OUR APPROACH: ADAGCN

### 2.1 ESTABLISHMENT OF ADAGCN

Consider an undirected graph $\mathcal{G} = (\mathcal{V}, \mathcal{E})$ with $N$ nodes $v_i \in \mathcal{V}$, edges $(v_i, v_j) \in \mathcal{E}$. $A \in \mathbb{R}^{N \times N}$ is the adjacency matrix with corresponding degree matrix $D_{ii} = \sum_j A_{ij}$. In the vanilla GCN model (Kipf & Welling, 2017) for semi-supervised node classification, the graph embedding of nodes with two convolutional layers is formulated as:

$$Z = \hat{A} \, \text{ReLU}(\hat{A} X W^{(0)}) W^{(1)} \tag{1}$$

where $Z \in \mathbb{R}^{N \times K}$ is the final embedding matrix (output logits) of nodes before softmax and $K$ is the number of classes. $X \in \mathbb{R}^{N \times C}$ denotes the feature matrix where $C$ is the input dimension. $\hat{A} = \tilde{D}^{-\frac{1}{2}} \tilde{A} \tilde{D}^{-\frac{1}{2}}$ where $\tilde{A} = A + I$ and $\tilde{D}$ is the degree matrix of $\tilde{A}$. In addition, $W^{(0)} \in \mathbb{R}^{C \times H}$ is the input-to-hidden weight matrix for a hidden layer with $H$ feature maps and $W^{(1)} \in \mathbb{R}^{H \times K}$ is the hidden-to-output weight matrix.

Our key motivation of constructing deep graph models is to efficiently explore information of high-order neighbors and then combine these messages from different orders of neighbors in an AdaBoost way. Nevertheless, if we naively extract information from high-order neighbors based on GCN, we are faced with stacking $l$ layers' parameter matrix $W^{(i)}, i = 0, ..., l-1$, which is definitely costly in computation. Besides, Multi-Scale Deep Graph Convolutional Networks (Luan et al., 2019) also theoretically demonstrated that the output can only contain the stationary information of graph structure and loses all the local information in nodes for being smoothed if we simply deepen GCN. Intuitively, the desirable representation of node features does not necessarily need too many nonlinear transformation $f$ applied on them. This is simply due to the fact that the feature of each node is normally one-dimensional sparse vector rather than multi-dimensional data structures, e.g., images, that intuitively need deep convolution network to extract high-level representation for vision tasks. This insight has been empirically demonstrated in many recent works (Wu et al., 2019; Klicpera et al., 2018; Xu et al., 2018a), showing that a two-layer fully-connected neural networks is a better choice in the implementation. Similarly, our AdaGCN also follows this direction by choosing an appropriate $f$ in each layer rather than directly deepen GCN layers.

Thus, we propose to remove ReLU to avoid the expensive joint optimization of multiple parameter matrices. Similarly, Simplified Graph Convolution (SGC) (Wu et al., 2019) also adopted this prac-

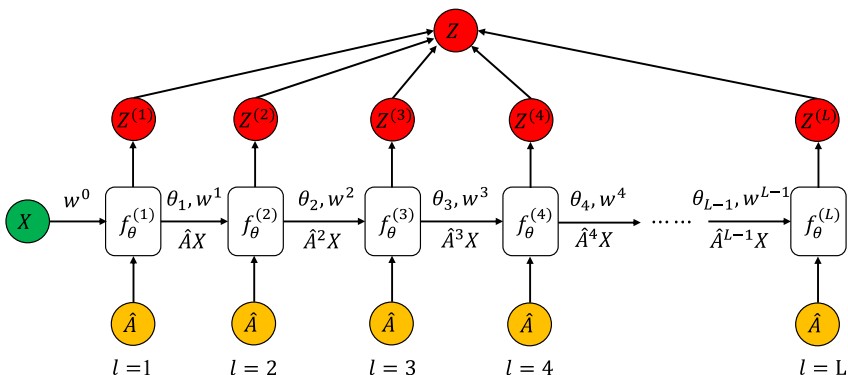

Figure 1: The RNN-like architecture of AdaGCN with each base classifier $f_\theta^{(l)}$ sharing the same neural network architecture $f_\theta$. $w^l$ and $\theta_l$ denote node weights and parameters computed after the $l$-th base classifier, respectively.

tice, arguing that nonlinearity between GCN layers is not crucial and the majority of the benefits arises from local weighting of neighboring features. Then the simplified graph convolution is:

$$Z = \hat{A}^l X W^{(0)} W^{(1)} \cdots W^{(l-1)} = \hat{A}^l X \tilde{W}, \tag{2}$$

where we collapse $W^{(0)} W^{(1)} \cdots W^{(l-1)}$ as $\tilde{W}$ and $\hat{A}^l$ denotes $\hat{A}$ to the $l$-th power. In particular, one crucial impact of ReLU in GCN is to accelerate the convergence of matrix multiplication since the ReLU is a contraction mapping intuitively. Thus, the removal of ReLU operation could also alleviate the oversmoothing issue, i.e. slowering the convergence of node embedding to indistinguishable ones (Li et al., 2018). Additionally, without ReLU this simplified graph convolution is also able to avoid the aforementioned joint optimization over multiple parameter matrices, resulting in computational benefits. Nevertheless, we find that this type of stacked linear transformation from graph convolution has *insufficient power in representing information of high-order neighbors*, which is revealed in our experiment described in Appendix A.2. Therefore, we propose to utilize an appropriate *nonlinear function* $f_\theta$, e.g., a two-layer fully-connected neural network, to replace the linear transformation $\tilde{W}$ in Eq. 2 and enhance the representation ability of *each base classifier* in AdaGCN as follows:

$$Z^{(l)} = f_\theta(\hat{A}^l X), \tag{3}$$

where $Z^{(l)}$ represents the final embedding matrix (output logits before Softmax) after the $l$-th base classifier in AdaGCN. This formulation also implies that the $l$-th base classifier in AdaGCN is extracting knowledge from features of current nodes and their $l$-th hop of neighbors. Due to the fact that the function of $l$-th base classifier in AdaGCN is similar to that of the $l$-th layer in other traditional GCN-based methods that directly stack many graph convolutional layers, *we regard the whole part of l-th base classifier as the l-th layers in AdaGCN*. As for the realization of Multi-class AdaBoost, we apply SAMME (Stagewise Additive Modeling using a Multi-class Exponential loss function) algorithm (Hastie et al., 2009), a natural and clean multi-class extension of the two-class AdaBoost adaptively combining weak classifiers.

As illustrated in Figure 1, we apply base classifier $f_\theta^{(l)}$ to extract knowledge from current node feature and $l$-th hop of neighbors by minimizing current weighted loss. Then we directly compute the weighted error rate $err^{(l)}$ and corresponding weight $\alpha^{(l)}$ of current base classifier $f_\theta^{(l)}$ as follows:

$$
\begin{aligned}
err^{(l)} &= \sum_{i=1}^{n} w_i \mathbb{I}\left(c_i \neq f_\theta^{(l)}(x_i)\right) / \sum_{i=1}^{n} w_i \\
\alpha^{(l)} &= \log \frac{1 - err^{(l)}}{err^{(l)}} + \log(K - 1),
\end{aligned}
\tag{4}
$$

where $w_i$ denotes the weight of $i$-th node and $c_i$ represents the category of current $i$-th node. To attain a positive $\alpha^{(l)}$, we only need $(1 - err^{(l)}) > 1/K$, i.e., the accuracy of each weak classifier

should be better than random guess (Hastie et al., 2009). This can be met easily to guarantee the weights to be updated in the right direction. Then we adjust nodes' weights by increasing weights on incorrectly classified ones:

$$w_i \leftarrow w_i \cdot \exp\left(\alpha^{(l)} \cdot \mathbb{I}\left(c_i \neq f_\theta^{(l)}(x_i)\right)\right), i = 1, \ldots, n \tag{5}$$

After re-normalizing the weights, we then compute $\hat{A}^{l+1}X = \hat{A} \cdot (\hat{A}^l X)$ to sequentially extract knowledge from $l$+1-th hop of neighbors in the following base classifier $f_\theta^{(l+1)}$. One crucial point of AdaGCN is that different from traditional AdaBoost, we only define one $f_\theta$, e.g. a two-layer fully connected neural network, *which in practice is recursively optimized in each base classifier just similar to a recurrent neural network*. This also indicates that the parameters from last base classifier are leveraged as the initialization of next base classifier, which coincides with our intuition that $l+1$-th hop of neighbors are directly connected from $l$-th hop of neighbors. The efficacy of this kind of layer-wise training has been similarly verified in (Belilovsky et al., 2018) recently. Further, we combine the predictions from different orders of neighbors in an Adaboost way to obtain the final prediction $C(A, X)$:

$$C(A, X) = \arg\max_k \sum_{l=0}^{L} \alpha^{(l)} f_\theta^{(l)}(\hat{A}^l X) \tag{6}$$

Finally, we obtain the concise form of AdaGCN in the following:

$$\begin{aligned}
\hat{A}^l X &= \hat{A} \cdot (\hat{A}^{l-1} X) \\
Z^{(l)} &= f_\theta^{(l)}(\hat{A}^l X) \\
Z &= \text{AdaBoost}(Z^{(l)})
\end{aligned} \tag{7}$$

Note that $f_\theta$ is non-linear, rather than linear in SGC (Wu et al., 2019), to guarantee the representation power. As shown in Figure 1, the architecture of AdaGCN is a variant of RNN with synchronous sequence input and output. *Although the same classifier architecture is adopted for $f_\theta^{(l)}$, their parameters are different, which is different from vanilla RNN*. We provide a detailed description of the our algorithm in Section 3.

## 2.2 COMPARISON WITH EXISTING METHODS

**Architectural Difference.** As illustrated in Figure 1 and 2, there is an apparent difference among the architectures of GCN (Kipf & Welling, 2017), SGC (Wu et al., 2019), Jumping Knowledge (JK) (Xu et al., 2018b) and AdaGCN. Compared with these existing graph convolutional approaches that sequentially convey intermediate result $Z^{(l)}$ to compute final prediction, our AdaGCN transmits weights of nodes $w^i$, aggregated features of different hops of neighbors $\hat{A}^l X$. More importantly, in AdaGCN the embedding $Z^{(l)}$ is independent of the flow of computation in the network and the sparse adjacent matrix $\hat{A}$ is also not directly involved in the computation of individual network because we compute

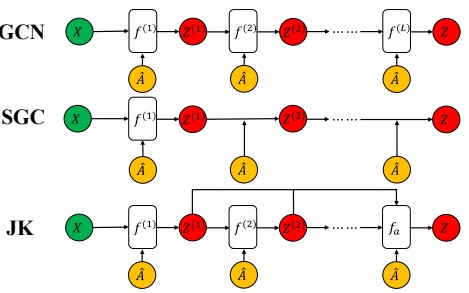

Figure 2: Comparison of the graph model architectures. $f_a$ in JK network denotes one aggregation layer with aggregation function such as concatenation or max pooling.

$\hat{A}^{(l+1)}X$ in advance and then feed it instead of $\hat{A}$ into the classifier $f_\theta^{(l+1)}$, thus yielding significant computation reduction, which will be discussed further in Section 3.

**Connection with PPNP and APPNP.** We also established a strong connection between AdaGCN and previous state-of-the-art Personalized Propagation of Neural Predictions (PPNP) and Approximate PPNP (APPNP) (Klicpera et al., 2018) method that leverages personalized pagerank to reconstruct graph convolutions in order to use information from a large and adjustable neighborhood. The analysis can be summarized in the following Proposition 1. Proof can refer to Appendix A.3.

**Proposition 1.** *Suppose that $\gamma$ is the teleport factor. Let matrix sequence $\{Z^{(l)}\}$ be from the output of each layer $l$ in AdaGCN, then PPNP is equivalent to the Exponential Moving Average (EMA) with exponentially decreasing factor $\gamma$ on $\{Z^{(l)}\}$ in a sharing parameters version, and its approximate version APPNP can be viewed as the approximated form of EMA with a limited number of terms.*

Proposition 1 illustrates that AdaGCN can be viewed as an *adaptive form* of APPNP, formulated as:

$$Z = \sum_{l=0}^{L} \alpha^{(l)} f_\theta^{(l)}(\hat{A}^l X) \tag{8}$$

Specifically, the first discrepancy between AdaGCN and APPNP lies in the adaptive coefficient $\alpha^{(l)}$ in AdaGCN determined by the error of $l$-th base classifier $f_\theta^{(l)}$ rather than fixed exponentially decreased weights in APPNP. In addition, AdaGCN employs classifier $f_\theta^{(l)}$ with different parameters to learn the embedding of different orders of neighbors, while APPNP shares these parameters in its form. We verified this benefit of our approach in our experiments shown in Section 4.2.

**Comparison with MixHop** MixHop (Abu-El-Haija et al., 2019) applied the similar way of graph convolution by repeatedly mixing feature representations of neighbors at various distance. Proposition 2 proves that both AdaGCN and MixHop are able to represent feature differences among neighbors while previous GCNs-based methods cannot. Proof can refer to Appendix A.4. Recap the definition of general layer-wise Neighborhood Mixing (Abu-El-Haija et al., 2019) as follows:

**Definition 1.** *General layer-wise Neighborhood Mixing: A graph convolution network has the ability to represent the layer-wise neighborhood mixing if for any $b_0, b_1, ..., b_L$, there exists an injective mapping $f$ with a setting of its parameters, such that the output of this graph convolution network can express the following formula:*

$$f\left(\sum_{l=0}^{L} b_l \sigma\left(\hat{A}^l X\right)\right) \tag{9}$$

**Proposition 2.** *AdaGCNs defined by our proposed approach (Eq. equation 7) are capable of representing general layer-wise neighborhood mixing, i.e., can meet the Definition 1.*

Albeit the similarity, AdaGCN distinguishes from MixHop in many aspects. Firstly, MixHop concatenates all outputs from each order of neighbors while we combines these predictions in an Adaboost way, which has theoretical generalization guarantee based on boosting theory Hastie et al. (2009). Oono & Suzuki (2020) have recently derived the optimization and generalization guarantees of multi-scale GNNs, serving as the theoretical backbone of AdaGCN. Meantime, MixHop allows full linear mixing of different orders of neighboring features, while AdaGCN utilizes different non-linear transformation $f_\theta^{(l)}$ among all layers, enjoying stronger expressive power.

## 3 ALGORITHM

In practice, we employ SAMME.R (Hastie et al., 2009), the soft version of SAMME, in AdaGCN. SAMME.R (R for Real) algorithm (Hastie et al., 2009) leverages real-valued confidence-rated predictions, i.e., weighted probability estimates, rather than predicted hard labels in SAMME, in the prediction combination, which has demonstrated a better generalization and faster convergence than SAMME. We elaborate the final version of AdaGCN in Algorithm 1. We provide the analysis on the choice of model depth $L$ in Appendix A.7, and then we elaborate the computational advantage of AdaGCN in the following.

**Analysis of Computational Advantage.** Due to the similarity of graph convolution in Mix-Hop (Abu-El-Haija et al., 2019), AdaGCN also requires no additional memory or computational complexity compared with previous GCN models. Meanwhile, our approach enjoys huge computational advantage compared with GCN-based models, e.g., PPNP and APPNP, stemming from excluding the additional computation involved in sparse tensors, such as the sparse tensor multiplication between $\hat{A}$ and other dense tensors, *in the forward and backward propagation of the neural network*. Specifically, there are only $L$ times sparse tensor operations for an AdaGCN model with $L$ layers, i.e., $\hat{A}^l X = \hat{A} \cdot (\hat{A}^{l-1} X)$ for each layer $l$. This operation in each layer yields a *dense tensor*

---

**Algorithm 1** AdaGCN based on SAMME.R Algorithm

---

**Input**: Features Matrix $X$, normalized adjacent matrix $\hat{A}$, a two-layer fully connected network $f_\theta$, number of layers $L$ and number of classes $K$.
**Output**: Final combined prediction $C(A, X)$.

1: Initialize the node weights $w_i = 1/n, i = 1, 2, ..., n$ on training set, neighbors feature matrix $\hat{X}^{(0)} = X$ and classifier $f_\theta^{(-1)}$.

2: **for** $l = 0$ to L **do**

3:     Fit the graph convolutional classifier $f_\theta^{(l)}$ on neighbor feature matrix $\hat{X}^{(l)}$ based on $f_\theta^{(l-1)}$ by minimizing current weighted loss.

4:     Obtain the weighted probability estimates $p^{(l)}(\hat{X}^{(l)})$ for $f_\theta^{(l)}$:
$$p_k^{(l)}(\hat{X}^{(l)}) = \text{Softmax}(f_\theta^{(l)}(c = k|\hat{X}^{(l)})), k = 1, \ldots, K$$

5:     Compute the individual prediction $h_k^{(l)}(x)$ for the current graph convolutional classifier $f_\theta^{(l)}$:
$$h_k^{(l)}(\hat{X}^{(l)}) \leftarrow (K-1) \left( \log p_k^{(l)}(\hat{X}^{(l)}) - \frac{1}{K} \sum_{k'} \log p_{k'}^{(l)}(\hat{X}^{(l)}) \right)$$
    where $k = 1, \ldots, K$.

6:     Adjust the node weights $w_i$ for each node $x_i$ with label $y_i$ on training set:
$$w_i \leftarrow w_i \cdot \exp \left( -\frac{K-1}{K} y_i^\top \log p^{(l)}(x_i) \right), i = 1, \ldots, n$$

7:     Re-normalize all weights $w_i$.

8:     Update $l$+1-hop neighbor feature matrix $\hat{X}^{(l+1)}$:
$$\hat{X}^{(l+1)} = \hat{A}\hat{X}^{(l)}$$

9: **end for**

10: Combine all predictions $h_k^{(l)}(\hat{X}^{(l)})$ for $l = 0, ..., L$.
$$C(A, X) = \arg\max_k \sum_{l=0}^{L} h_k^{(l)}(\hat{X}^{(l)})$$

11: **return** Final combined prediction $C(A, X)$.

---

$B^l = \hat{A}^l X$ for the $l$-th layer, which is then fed into the computation in a two-layer fully-connected network, i.e., $f_\theta^{(l)}(B^l) = \text{ReLU}(B^l W^{(0)}) W^{(1)}$. Due to the fact that dense tensor $B^l$ has been computed in advance, there is no other computation related to sparse tensors in the multiple forward and backward propagation procedures while training the neural network. By contrast, this multiple computation involved in sparse tensors in the GCN-based models, e.g., GCN: $\hat{A} \text{ReLU}(\hat{A} X W^{(0)}) W^{(1)}$, is highly expensive. AdaGCN avoids these additional sparse tensor operations in the neural network and then attains huge computational efficiency. We demonstrate this viewpoint in the Section 4.3.

## 4 EXPERIMENTS

**Experimental Setup.** We select five commonly used graphs: CiteSeer, Cora-ML (Bojchevski & Günnemann, 2018; McCallum et al., 2000), PubMed (Sen et al., 2008), MS-Academic (Shchur et al., 2018) and Reddit. Dateset statistics are summarized in Table 1. Recent graph neural networks suffer from overfitting to a single splitting of training, validation and test datasets (Klicpera et al., 2018). To address this problem, inspired by (Klicpera et al., 2018), we test all approaches on multiple random splits and initialization to conduct a rigorous study. Detailed dataset splittings are provided in Appendix A.6.

| Dateset | Nodes | Edges | Classes | Features | Label Rate |
|---|---|---|---|---|---|
| CiteSeer | 3,327 | 4,732 | 6 | 3,703 | 3.6% |
| Cora | 2,708 | 5,429 | 7 | 1,433 | 5.2% |
| PubMed | 19,717 | 44,338 | 3 | 500 | 0.3% |
| MS Academic | 18,333 | 81,894 | 15 | 6,805 | 1.6% |
| Reddit | 232,965 | 11,606,919 | 41 | 602 | 65.9% |

Table 1: Dateset statistics

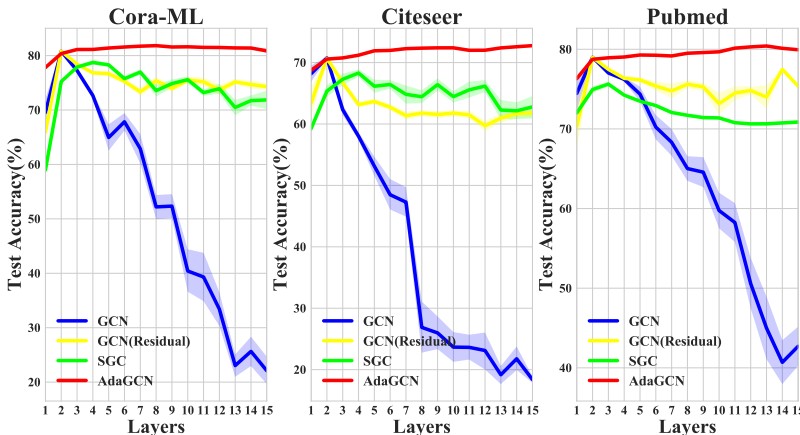

Figure 3: Comparison of test accuracy of different models as the layer increases. We regard the $l$-th base classifier as the $l$-th layer in AdaGCN as both of them are leveraged to exploit the information from $l$-th order of neighbors for current nodes.

**Basic Setting of Baselines and AdaGCN.** We compare AdaGCN with GCN (Kipf & Welling, 2017) and Simple Graph Convolution (SGC) (Wu et al., 2019) in Figure 3. In Table 2, we employ the same baselines as (Klicpera et al., 2018): V.GCN (vanilla GCN) (Kipf & Welling, 2017) and GCN with our early stopping, N-GCN (network of GCN) (Abu-El-Haija et al., 2018a), GAT (Graph Attention Networks) (Veličković et al., 2018), BT.FP (bootstrapped feature propagation) (Buchnik & Cohen, 2018) and JK (jumping knowledge networks with concatenation) (Xu et al., 2018b). In the computation part, we additionally compare AdaGCN with FastGCN (Chen et al., 2018) and GraphSAGE (Hamilton et al., 2017). We refer to the result of baselines from (Klicpera et al., 2018) and the implementation of AdaGCN is adapted from APPNP. For AdaGCN, after the line search on hyper-parameters, we set $h = 5000$ hidden units for the first four datasets except Ms-academic with $h = 3000$, and 15, 12, 20 and 5 layers respectively due to the different graph structures. In addition, we set dropout rate to 0 for Citeseer and Cora-ML datasets and 0.2 for the other datasets and $5 \times 10^{-3} L_2$ regularization on the first linear layer. We set weight decay as $1 \times 10^{-3}$ for Citeseer while $1 \times 10^{-4}$ for others. More detailed model parameters and analysis about our early stopping mechanism can be referred from Appendix A.6.

## 4.1 Design of Deep Graph Models to Circumvent Oversmoothing Effect

It is well-known that GCN suffers from oversmoothing (Li et al., 2018) with the stacking of more graph convolutions. However, combination of knowledge from each layer to design deep graph

| Model | Citeseer | Cora-ML | Pubmed | MS Academic |
|---|---|---|---|---|
| V.GCN | 73.51±0.48 | 82.30±0.34 | 77.65±0.40 | 91.65±0.09 |
| GCN | 75.40±0.30 | 83.41±0.39 | 78.68±0.38 | 92.10±0.08 |
| N-GCN | 74.25±0.40 | 82.25±0.30 | 77.43±0.42 | 92.86±0.11 |
| GAT | 75.39±0.27 | 84.37±0.24 | 77.76±0.44 | 91.22±0.07 |
| JK | 73.03±0.47 | 82.69±0.35 | 77.88±0.38 | 91.71±0.10 |
| BT.FP | 73.55±0.57 | 80.84±0.97 | 72.94±1.00 | 91.61±0.24 |
| PPNP | 75.83±0.27 | 85.29±0.25 | OOM | OOM |
| APPNP | 75.73±0.30 | 85.09±0.25 | 79.73±0.31 | 93.27±0.08 |
| PPNP (ours) | 75.53±0.32 | 84.39±0.28 | OOM | OOM |
| APPNP (ours) | 75.41±0.35 | 84.28±0.28 | 79.41±0.34 | 92.98±0.07 |
| AdaGCN | **76.68±0.20** | **85.97±0.20** | **79.95±0.21** | **93.17±0.07** |
| P value | $1.8 \times 10^{-15}$ | $2.2 \times 10^{-16}$ | $1.1 \times 10^{-5}$ | $2.1 \times 10^{-9}$ |

Table 2: Average accuracy under 100 runs with uncertainties showing the 95 % confidence level calculated by bootstrapping. OOM denotes "out of memory". "(ours)" denotes the results based on our implementation, which are slight lower than numbers above from original literature (Klicpera et al., 2018). P values of paired t test between APPNP (ours) and AdaGCN are provided in the last row.

|  | Citeseer | Cora-ML | Pubmed | MS Academic |
|---|---|---|---|---|
| **Label Rates** | 1.0% / 2.0% | 2.0% / 4.0% | 0.1% / 0.2% | 0.6% / 1.2% |
| V.GCN | 67.6±1.4/70.8±1.4 | 76.4±1.3/81.7±0.8 | 70.1±1.4/74.6±1.6 | 89.7±0.4/91.1±0.2 |
| GCN | 70.3±0.9/72.7±1.1 | 80.0±0.7/82.8±0.9 | 71.1±1.1/75.2±1.0 | 89.8±0.4/91.2±0.3 |
| PPNP | 72.5±0.9/74.7±0.7 | 80.1±0.7/83.0±0.6 | OOM | OOM |
| APPNP | 72.2±1.3/74.2±1.1 | 80.1±0.7/83.2±0.6 | 74.0±1.5/77.2±1.2 | 91.7±0.2/92.6±0.2 |
| AdaGCN | **74.2±0.3/75.5±0.3** | **83.7±0.3/85.3±0.2** | **77.1±0.5/79.3±0.3** | **92.1±0.1/92.7±0.1** |

Table 3: Average accuracy across different label rates with 20 splittings of datasets under 100 runs.

models is a reasonable method to circumvent oversmoothing issue. In our experiment, we aim to explore the prediction performance of GCN, GCN with residual connection (Kipf & Welling, 2017), SGC and our AdaGCN with a growing number of layers.

From Figure 3, it can be easily observed that oversmoothing leads to the rapid decreasing of accuracy for GCN (blue line) as the layer increases. In contrast, the speed of smoothing (green line) of SGC is much slower than GCN due to the lack of ReLU analyzed in Section 2.1. Similarly, GCN with residual connection (yellow line) partially mitigates the oversmoothing effect of original GCN but fails to take advantage of information from different orders of neighbors to improve the prediction performance constantly. Remarkably, AdaGCN (red line) is able to consistently enhance the performance with the increasing of layers across the three datasets. This implies that AdaGCN can efficiently incorporate knowledge from different orders of neighbors and circumvent oversmoothing of original GCN in the process of constructing deep graph models. In addition, the fluctuation of performance for AdaGCN is much lower than GCN especially when the number of layer is large.

## 4.2 PREDICTION PERFORMANCE

We conduct a rigorous study of AdaGCN on four datasets under multiple splittings of dataset. The results from Table 2 suggest the state-of-the-art performance of our approach and the improvement compared with APPNP validates the benefit of adaptive form for our AdaGCN. More rigorously, p values under paired t test demonstrate the significance of improvement for our method.

In the realistic setting, graphs usually have different labeled nodes and thus it is necessary to investigate the robust performance of methods on different number of labeled nodes. Here we utilize label rates to measure the different numbers of labeled nodes and then sample corresponding labeled nodes per class on graphs respectively. Table 3 presents the consistent state-of-the-art performance of AdaGCN under different label rates. An interesting manifestation from Table 3 is that AdaGCN yields more improvement on fewer label rates compared with APPNP, showing more efficiency on graphs with few labeled nodes. Inspired by the Layer Effect on graphs (Sun et al., 2019), we argue that the increase of layers in AdaGCN can result in more benefits on the efficient propagation of label signals especially on graphs with limited labeled nodes.

More rigorously, we additionally conduct the comparison on a larger dataset, i.e., Reddit. We choose the best layer as 4 due to the fact that AdaGCN with larger number of layers tends to suffer from overfitting on this relatively simple dataset (with high label rate 65.9%). Table 4 suggests that AdaGCN can still outperform other typical baselines, including V.GCN, PPNP and APPNP. More experimental details can be referred from Appendix A.6.

| Reddit | F1-Score | Per-epoch training time |
|---|---|---|
| V.GCN | 94.46±0.06 | 5627.46ms |
| PPNP | OOM | OOM |
| APPNP | 95.04±0.07 | 29489.81ms |
| AdaGCN | **95.39±0.13** | **32.29ms** |

Table 4: Average F1-scores and per-epoch training time of typical methods on Reddit dataset under 5 runs.

## 4.3 COMPUTATIONAL EFFICIENCY

Without the additional computational cost involved in sparse tensors in the propagation of the neural network, AdaGCN presents huge computational efficiency. From the left part of Figure 4, it exhibits that AdaGCN has the fastest speed of per-epoch training time in comparison with other methods except the comparative performance with FastGCN in Pubmed. In addition, there is a somewhat inconsistency in computation of FastGCN, with fastest speed in Pubmed but slower than

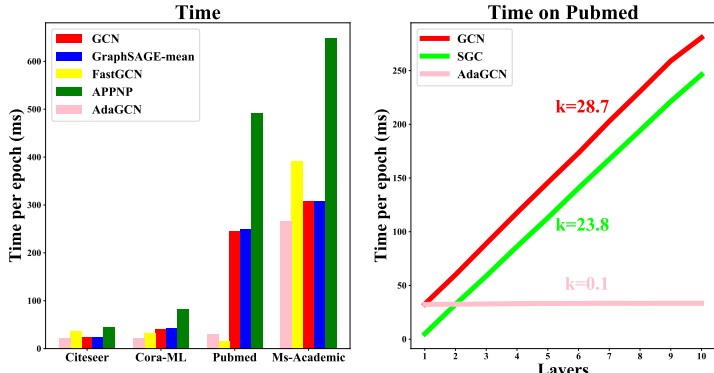

Figure 4: Left: Per-epoch training time of AdaGCN vs other methods under 5 runs on four datasets. Right: Per-epoch training time of AdaGCN compared with GCN and SGC with the increasing of layers and the digit after "$k =$" denotes the slope in a fitted linear regression.

GCN on Cora-ML and MS-Academic datasets. Furthermore, with multiple power iterations involved in sparse tensors, APPNP unfortunately has relatively expensive computation cost. It should be noted that this computational advantage of AdaGCN is more significant when it comes to large datasets, e.g., Reddit. Table 4 demonstrates AdaGCN has the potential to perform much faster on larger datasets.

Besides, we explore the computational cost of ReLU and sparse adjacency tensor with respect to the number of layers in the right part of Figure 4. We focus on comparing AdaGCN with SGC and GCN as other GCN-based methods, such as GraphSAGE and APPNP, behave similarly with GCN. Particularly, we can easily observe that both SGC (green line) and GCN (red line) show a linear increasing tendency and GCN yields a larger slope arises from ReLU and more parameters. For SGC, stacking more layers directly is undesirable regarding the computation. Thus, a limited number of SGC layers is preferable with more advanced optimization techniques Wu et al. (2019). It also shows that the computational cost involved sparse matrices in neural networks plays a dominant role in all the cost especially when the layer is large enough. In contrast, our AdaGCN (pink line) displays an almost constant trend as the layer increases simply because it excludes the extra computation involved in sparse tensors $\hat{A}$, such as $\cdots \hat{A} \text{ReLU}(\hat{A}XW^{(0)})W^{(1)} \cdots$, in the process of training neural networks. AdaGCN maintains the updating of parameters in the $f_\theta^{(l)}$ *with a fixed architecture* in each layer while the layer-wise optimization, therefore displaying a nearly constant computation cost within each epoch although more epochs are normally needed in the entire layer-wise training. We leave the analysis of exact time and memory complexity of AdaGCN as future works, but boosting-based algorithms including AdaGCN is memory-efficient (Oono & Suzuki, 2020).

## 5    DISCUSSIONS AND CONCLUSION

One potential concern is that AdaBoost (Hastie et al., 2009; Freund et al., 1999) is established on i.i.d. hypothesis while graphs have inherent data-dependent property. Fortunately, the statistical convergence and consistency of boosting (Lugosi & Vayatis, 2001; Mannor et al., 2003) can still be preserved when the samples are weakly dependent (Lozano et al., 2013). More discussion can refer to Appendix A.5. In this paper, we propose a novel RNN-like deep graph neural network architecture called AdaGCNs. With the delicate architecture design, our approach AdaGCN can effectively explore and exploit knowledge from different orders of neighbors in an Adaboost way. Our work paves a way towards better combining different-order neighbors to design deep graph models rather than only stacking on specific type of graph convolution.

### ACKNOWLEDGMENTS

Z. Lin is supported by NSF China (grant no.s 61625301 and 61731018), Major Scientific Research Project of Zhejiang Lab (grant no.s 2019KB0AC01 and 2019KB0AB02), Beijing Academy of Artificial Intelligence, and Qualcomm.

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

# A  APPENDIX

## A.1  RELATED WORKS ON DEEP GRAPH MODELS

A straightforward solution (Kipf & Welling, 2017; Xu et al., 2018b) inspired by ResNets (He et al., 2016) was by adding residual connections, but this practice was unsatisfactory both in prediction performance and computational efficiency towards building deep graph models, as shown in our experiments in Section 4.1 and 4.3. More recently, JK (Jumping Knowledge Networks (Xu et al., 2018b)) introduced jumping connections into final aggregation mechanism in order to extract knowledge from different layers of graph convolutions. However, this straightforward change of GCN architecture exhibited inconsistent empirical performance for different aggregation operators, which cannot demonstrate the successful construction of deep layers. In addition, Graph powering-based method (Jin et al., 2019) implicitly leveraged more spatial information by extending classical spectral graph theory to robust graph theory, but they concentrated on defending adversarial attacks rather than model depth. LanczosNet (Liao et al., 2019) utilized Lanczos algorithm to construct low rank approximations of the graph Laplacian and then can exploit multi-scale information. Moreover, APPNP (Approximate Personalized Propagation of Neural Predictions, (Klicpera et al., 2018)) leveraged the relationship between GCN and personalized PageRank to derive an improved global propagation scheme. Beyond these, DeepGCNs (Li et al., 2019) directly adapted residual, dense connection and dilated convolutions to GCN architecture, but it mainly focused on the task of point cloud semantic segmentation and has not demonstrated its effectiveness in typical graph tasks. Similar to our work, Deep Adaptive Graph Neural Network (DAGNN) (Liu et al., 2020) also focused on incorporating information from large receptive fields through the entanglement of representation transformation and propagation, while our work efficiently ensembles knowledge from large receptive fields in an Adaboost manner. Other related works based on global attention models (Puny et al., 2020) and sample-based methods (Zeng et al., 2019) are also helpful to construct deep graph models.

## A.2  INSUFFICIENT REPRESENTATION POWER OF ADASGC

As illustrated in Figure 5, with the increasing of layers, AdaSGC with only linear transformation has insufficient representation power both in extracting knowledge from high-order neighbors and combining information from different orders of neighbors while AdaGCN exhibits a consistent improvement of performance as the layer increases.

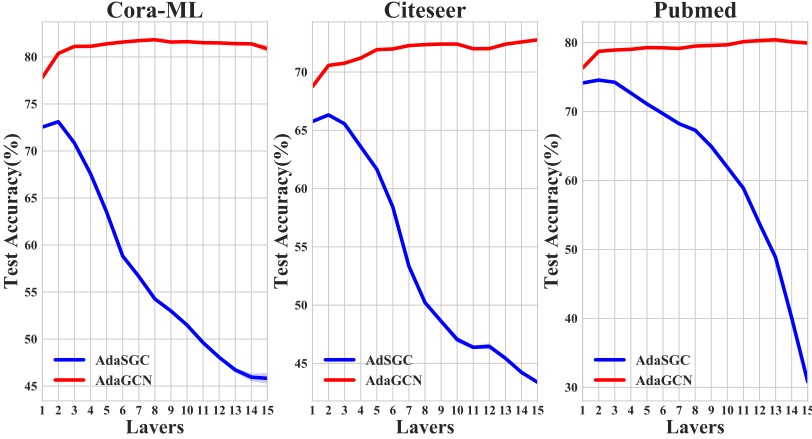

Figure 5: AdaSGC vs AdaGCN.

## A.3  PROOF OF PROPOSITION 1

Firstly, we further elaborate the Proposition 1 as follows, then we provide the proof.

Suppose that $\gamma$ is the teleport factor. Consider the output $Z_{\text{PPNP}} = \gamma(\mathbb{I} - (1 - \gamma)\hat{A})^{-1} f_\theta(X)$ in PPNP and $Z_{\text{APPNP}}$ from its approxminated version APPNP. Let matrix sequence $\{Z^{(l)}\}$ be from the output of each layer $l$ in AdaGCN, then PPNP is equivalent to the Exponential Moving Average (EMA) with exponentially decreasing factor $\gamma$, a first-order infinite impulse response filter, on $\{Z^{(l)}\}$ in a sharing parameters version, i.e., $f_\theta^{(l)} \equiv f_\theta$. In addition, APPNP, which we reformulate in Eq. 10, can be viewed as the approximated form of EMA with a

limited number of terms.

$$Z_{\text{APPNP}} = (\gamma \sum_{l=0}^{L-1} (1-\gamma)^l \hat{A}^l + (1-\gamma)^L \hat{A}^L) f_\theta(X) \tag{10}$$

*Proof.* According to Neumann Theorem, $Z_{\text{PPNP}}$ can be expanded as a Neumann series:

$$Z_{\text{PPNP}} = \gamma(\mathbb{I} - (1-\gamma)\hat{A})^{-1} f_\theta(X)$$

$$= \gamma \sum_{l=0}^{\infty} (1-\gamma)^l \hat{A}^l f_\theta(X),$$

where feature embedding matrix sequence $\{Z^{(l)}\}$ for each order of neighbors share the same parameters $f_\theta$. If we relax this sharing nature to the adaptive form with respect to the layer and put $\hat{A}^l$ into $f_\theta$, then the output $Z$ can be approximately formulated as:

$$Z_{\text{PPNP}} \approx \gamma \sum_{l=0}^{\infty} (1-\gamma)^l f_\theta^{(l)}(\hat{A}^l X)$$

This relaxed version from PPNP is the Exponential Moving Average form of matrix sequence $\{Z^{(l)}\}$ with exponential decreasing factor $\gamma$. Moreover, if we approximate the EMA by truncating it after $L-1$ items, then the weight omitted by stopping after $L-1$ items is $(1-\gamma)^L$. Thus, the approximated EMA is exactly the APPNP form:

$$Z_{\text{APPNP}} = (\gamma \sum_{l=0}^{L-1} (1-\gamma)^l \hat{A}^l + (1-\gamma)^L \hat{A}^L) f_\theta(X)$$

$\square$

## A.4 PROOF OF PROPOSITION 2

*Proof.* We consider a two layers fully-connected neural network as $f$ in Eq. 8, then the output of AdaGCN can be formulated as:

$$Z = \sum_{l=0}^{L} \alpha^{(l)} \sigma(\hat{A}^l X W^{(0)}) W^{(1)}$$

Particularly, we set $W^{(0)} = \frac{b_l}{\text{sign}(b_l)\alpha^{(l)}} \mathbb{I}$ and $W^{(1)} = \text{sign}(b_l)\mathbb{I}$ where $\text{sign}(b_l)$ is the signed incidence scalar w.r.t $b_l$. Then the output of AdaGCN can be presented as:

$$Z = \sum_{l=0}^{L} \alpha^{(l)} \sigma(\hat{A}^l X \frac{b_l}{\text{sign}(b_l)\alpha^{(l)}} \mathbb{I})\text{sign}(b_l)\mathbb{I}$$

$$= \sum_{l=0}^{L} \alpha^{(l)} \sigma(\hat{A}^l X) \frac{b_l}{\text{sign}(b_l)\alpha^{(l)}} \text{sign}(b_l)$$

$$= \sum_{l=0}^{L} b_l \sigma\left(\hat{A}^l X\right)$$

The proof that GCNs-based methods are not capable of representing general layer-wise neighborhood mixing has been demonstrated in MixHop (Abu-El-Haija et al., 2019). Proposition 2 proved. $\square$

## A.5 EXPLANATION ABOUT CONSISTENCY OF BOOSTING ON DEPENDENT DATA

**Definition 2.** *($\beta$-mixing sequences.) Let $\sigma_i^j = \sigma(W) = \sigma(W_i, W_{i+1}, ..., W_j)$ be the $\sigma$-field generated by a strictly stationary sequence of random variables $W = (W_i, W_{i+1}, ..., W_j)$. The $\beta$-mixing coefficient is defined by:*

$$\beta_W(n) = \sup_k \mathbb{E} \sup \left\{ \left| \mathbb{P}\left(A|\sigma_1^k\right) - \mathbb{P}(A) \right| : A \in \sigma_{k+n}^\infty \right\}$$

*Then a sequence $W$ is called $\beta$-mixing if $\lim_{n \to \infty} \beta_W(n) = 0$. Further, it is algebraically $\beta$-mixing if there is a positive constant $r_\beta$ such that $\beta_W(n) = \mathcal{O}(n^{-r_\beta})$.*

**Definition 3.** *(Consistency) A classification rule is consistent for a certain distribution $P$ if $E(L(h_n)) = P\{h_n(X) = Y\} \to a$ as $n \to \infty$ where $a$ is a constant. It is strongly Bayes-risk consistent if $\lim_{n \to \infty} L(h_n) = a$ almost surely.*

Under these definitions, the convergence and consistence of regularized boosting method on stationary $\beta$-mixing sequences can be proved under mild assumptions. More details can be referred from (Lozano et al., 2013).

## A.6    Experimental Details

**Early Stopping on AdaGCN.** We apply the same early stopping mechanism across all the methods as (Klicpera et al., 2018) for fair comparison. Furthermore, boosting theory also has the capacity to perfectly incorporate early stopping and it has been shown that for several boosting algorithms including AdaBoost, this regularization via early stopping can provide guarantees of consistency (Zhang et al., 2005; Jiang et al., 2004; Bühlmann & Yu, 2003).

**Dataset Splitting.** We choose a training set of a fixed nodes per class, an early stopping set of 500 nodes and test set of remained nodes. Each experiment is run with 5 random initialization on each data split, leading to a total of 100 runs per experiment. On a standard setting, we randomly select 20 nodes per class. For the two different label rates on each graph, we select 6, 11 nodes per class on citeseer, 8, 16 nodes per class on Cora-ML, 7, 14 nodes per class on Pubmed and 8, 15 nodes per class on MS-Academic dataset.

**Model parameters.** For all GCN-based approaches, we use the same hyper-parameters in the original paper: learning rate of 0.01, 0.5 dropout rate, $5 \times 10^{-4}$ $L_2$ regularization weight, and 16 hidden units. For FastGCN, we adopt the officially released code to conduct our experiments. PPNP and APPNP are adapted with best setting: $K = 10$ power iteration steps for APPNP, teleport probability $\gamma = 0.1$ on Cora-ML, Citeseer and Pubmed, $\gamma = 0.2$ on Ms-Academic. In addition, we use two layers with $h = 64$ hidden units and apply L2 regularization with $\lambda = 5 \times 10^{-3}$ on the weights of the first layer and use dropout with dropout rate $d = 0.5$ on both layers and the adjacency matrix. The early stopping criterion uses a patience of $p = 100$ and an (unreachably high) maximum of $n = 10000$ epochs.The implementation of AdaGCN is adapted from PPNP and APPNP. Corresponding patience $p = 300$ and $n = 500$ in the early stopping of AdaGCN. Moreover, SGC is re-implemented in a straightforward way without incorporating advanced optimization for better illustration and comparison. Other baselines are adopted the same parameters described in PPNP and APPNP.

**Settings on Reddit dataset.** By repeatedly tuning the parameters of these typical methods on Reddit, we finally choose weight decay rate as $10^{-4}$, hidden layer size 100 and epoch 20000 for AdaGCN. For APPNP, we opt weight decay rate as $10^{-5}$, dropout rate as 0 and epoch 500. V.GCN applies the same parameters in (Kipf & Welling, 2017) and we choose epoch as 500. All approaches have not deployed early stopping due to the expensive computational cost on the large Reddit dataset, which is also a fair comparison.

## A.7    Choice of the Number of Layers

Different from the "forcible" behaviors in CNNs that directly stack many convolution layers, in our AdaGCN there is a theoretical guidance on the choice of model depth $L$, i.e., the number of base classifiers or layers, derived from boosting theory. Specifically, according to the boosting theory, the increasing of $L$ can exponentially decreases the empirical loss, however, from the perspective of VC-dimension, an overly large $L$ can yield overfitting of AdaGCN. It should be noted that the deeper graph convolution layers in AdaGCN are not always better, which indeed heavily depends on the the complexity of data. In practice, $L$ can be determined via cross-validation. Specifically, we start a VC-dimension-based analysis to illustrate that too large $L$ can yield overfitting of AdaGCN. For $L$ layers of AdaGCN, its hypothesis set is

$$\mathcal{F}_L = \left\{ \arg\max_k \left( \sum_{l=1}^{L} \alpha^{(l)} f_\theta^{(l)} \right) : \alpha^{(l)} \in \mathbb{R}, l \in [1, L] \right\} \tag{11}$$

Then the VC-dimension of $\mathcal{F}_T$ can be bounded as follows in terms of the VC-dimension $d$ of the family of base hypothesis:

$$\text{VCdim}(\mathcal{F}_L) \leq 2(d+1)(L+1)\log_2((L+1)e), \tag{12}$$

where $e$ is a constant and the upper bounds grows as $L$ increases. Combined with VC-dimension generalization bounds, these results imply that larger values of $L$ can lead to overfitting of AdaBoost. This situation also happens in AdaGCN, which inspires us that there is no need to stack too many layers on AdaGCN in order to avoid overfitting. In practice, $L$ is typically determined via cross-validation.

