# OpenReview forum: "AdaGCN: Adaboosting Graph Convolutional Networks into Deep Models"
_ICLR.cc/2021/Conference — ICLR 2021 Poster_

### Official Review · AnonReviewer3 · 2020-10-27
**AdaGCN: Adaboosting Graph Convolutional Networks Into Deep Models**

**Rating:** 5
**Confidence:** 4

**Review:**

By integrating Adaboosting and a fully connected layer, this paper provides a new graph neural network structure. The objective of this paper is to design a deeper graph models in an efficient way for better performance. The computational efficiency and performance of the proposed algorithm are evaluated using the task of node property prediction on several public datasets. This is a new variant of GNN, but the quality this paper is lower than the expectation regarding to the clarity and organisation.

Pros:
1.	The algorithm integrated Adaboosting for graph data. Thus, AdaGCN could utilise different levels of node features for final prediction.
2.	The method is optimised in a layer-wise way rather than the traditional GCN optimisation, which is similar to the optimisation of recurrent neural networks.
3.	Authors compared the structure of AdaGCN with that of other GNN variants.
4.	For the experiments, the proposed algorithm is more computationally efficient, and achieves better performance on the task of node property prediction.  The performance of AdaGCN is slightly more robust than previous methods. The performance drop is not observed within 10 layers for AdaGCN as shown in Fig 3.

Cons:
1.	Speaking of the state of art performance, GraphSAGE with LSTM also achieves a 95.4% F1 score on the Reddit dataset for node classification tasks. Thus, the authors may need to compare the training time and performance with more recent algorithms, like ClusterGCN and GraphSAGE.
2.	The paper is not well written. Many typos are discovered. For example, extra space is added in the first sentence after equation 3. Meanwhile, the punctuation around equations is not consistent. For the full sentence following an equation, one would place a full stop after the equation. However, there is no full stop after equations 5, 6, 7, and 8. Abbreviations, such as "JK", "APPNP", and "PPNP", are used before introduction.
3.	Some notations are confusing and misleading. $K$ refers to the number of node categories, and $k$ refers to a category of a node. Meanwhile, $w_i$ and $W^{l}$ have completely irrelevant definitions.
4.	To evaluate the efficiency of different GCN approaches, the authors listed the per-epoch training time of methods. The implementation of GCN with different frameworks would result in the large variance of training time. It is better that the authors could include the time and memory complexity of each algorithm.
5.	Fig 4, after 10 layers, it is not clear whether the linear trend would continue. This result is a bit misleading.

---

> ### Author Response · Authors · 2020-11-16
> **Response to Reviewer 3**
>
> Thank you for your constructive feedback. We answer your questions below:
>
> 1. Reddit dataset.
>
> As stated on page 8 of our paper, Reddit is a relatively simple dataset as it has a high label rate (65.9%). Thus, it can be understood that some previous methods can still achieve similar performance on this simple dataset. However, from the comparison of computation cost shown in the left part of Figure 4, we observed that GraphSAGE-mean has a comparable computation cost with GCN. Plus, the huge advantage of per-epoch training time of AdaGCN compared with GCN as shown in Table 4, can provide sufficient empirical proof to demonstrate that AdaGCN still significantly outperforms Graph-SAGE and similar methods in terms of computation cost.
>
> 2. Writing.
>
> Thank you for this suggestion about writing. We promise to correct typos and improve the presentation in the revised version.
>
> 3. Notations.
>
> When $K$ refers to the number of classes, it is natural to leverage $k$ to refer to the specific category of a node. Besides, $w_i$ indicates the weight of node $i$ while $w^l$ represents the all weight matrix of nodes in the $l$-th base classifier, for which we used the superscript for the distinguishment. We will make the definitions of notations clearer.
>
> 4. Time complexity.
>
> In our experiments, we evaluate the efficiency of different GCN approaches under the same framework while observing the limited variance of training time. We will add the analysis of time and memory complexity if accepted.
>
> 5. Figure 4.
>
> The nearly constant tendency stems from the computational advantage of AdaGCN, for which we conducted a rigorous analysis in Section 3. Concretely speaking, our RNN-like architecture avoids the additional computation related to sparse matrices A in a network. Thus, the linear tendency will continue even after 10 layers.

---

### Official Review · AnonReviewer1 · 2020-10-29
**This paper proposed a novel RNN-like deep graph neural network architecture by incorporating AdaBoost into the computation of network for extracting the knowledge from high-order local neighborhood.**

**Rating:** 6
**Confidence:** 5

**Review:**

Overall, the proposed AdaGCN model could incorporate the different hops of neighbors into the network in an Adaboost way without improving the computational cost. That has been confirmed by the theoretical comparison with other baselines and the experimental results.

Pros:
[1] It proposed a novel deep graph neural network by incorporating AdaBoost into the computation of network.
[2] It compared to the existing related work to illustrate the benefits of the proposed AdaGCN.
[3] The experiments demonstrate its effectiveness of efficiency of AdaGCN on encoding the high-order graph structure information.

Cons:
[1] The simplified graph convolution might be vulnerable to the noisy nodes. That is, when there exists one noisy node with abnormal attributes, this simplified graph convolution might be significantly degraded. Thus, the higher-order convolution in the proposed method might become worse.
[2] In table 2, it is confusing why the implemented methods (e.g., PPNP (ours), APPNP (ours)) have lower performance than the results reported in the literature (Klicpera et al., 2018).
[3] In Section 4.3, it shows that Fast-GCN has fastest speed in Pubmed, but slower than GCN on Cora-ML and MS-Academic datasets. That might need more explanation since intuitively the goal of Fast-GCN is to improve the efficiency of GCN.

---

> ### Author Response · Authors · 2020-11-16
> **Response to Reviewer 1**
>
> We thank you for your recommendation for acceptance. Here is our clarification.
>
> [1] Noisy nodes
>
> Robustness against noisy nodes is an interesting issue. However, we argue that the issue you mentioned may not happen in our AdaGCN, mainly due to the fact that we have employed the early stopping mechanism. During the layer-wise optimization, if higher-order graph convolution achieves worse performance, early stopping, which serves as a regularization, can help avoid this performance degradation and finally selects a model with a relatively small number of layers. We will leave this issue as the future work to explore.
>
> [2] Table 2
>
> We argue that they are sufficiently close although there is a slight gap. Randomness may be part of the reasons.
>
> [3] Fast-GCN
>
> We conjecture that it is linked with the sensitivity of hyper-parameters about important sampling in FastGCN. Specifically, our preprocessing of the graphs, which followed Klicpera et al., 2018, is slightly different from FastGCN, and for convenience, we still used the default optimal hyper-parameters of FastGCN for each dataset, respectively. Thus, the sensitivity of sampling hyper-parameters in FastGCN might cause a slight decreasing in its efficiency on some datasets. Overall, we can still admit the efficacy of FastGCN, but we are expected to carefully tune its hyper-parameters of sampling in the practice. Moreover, it is also safe to claim that our AdaGCN is competitive in computation efficiency.

---

### Official Review · AnonReviewer2 · 2020-10-31
**Adaboosting GCNs**

**Rating:** 7
**Confidence:** 3

**Review:**

Summary
In this paper, the authors study graph convolutional networks, where they propose to use AdaBoost for Deep GCNs. This method makes it possible to use information from multi-hop neighbours. Computationally, the proposed method is efficient, which is illustrated through various experiments. The paper is well written with good clarity, while the proposed method is novel and significant to the research community.

Reasons for recommending acceptance
- To that of reviewer's knowledge, the proposed scheme is novel. The method address the issue of using information for higher-order neighbours, without increasing the computational complexity.
- comprehensive experiments across multiple datasets, evaluating AdaGCN in terms of computational efficiency, accuracy, dependency on the number of layers.

Questions
- While comparing MixHop against AdaGCN, the authors mention that AdaGCN does have generalization guarantees from Boosting theory. This statement is loose, and a formal justification may be needed.
- In Figure 4 (Right), where the epoch time is measured against the number of layers, AdaGCN is shown to have nearly constant time w.r.t. layers. Some more explanation would be useful in understanding this.

---

> ### Author Response · Authors · 2020-11-16
> **Response to Reviewer 2**
>
> We thank you for your recommendation for acceptance. Here is our response.
>
> 1. Generalization guarantee.
>
> Thank you for this suggestion. We will consider adding a formal justification for the generalization guarantee of our method derived from the boosting theory in the revised version.
>
> 2. Figure 4.
>
> The nearly constant tendency stems from the computational advantage of AdaGCN, which we analyzed in Section 3. Concretely speaking, our special RNN-like architecture avoids the additional computation related to sparse matrices A in a network. At your suggestion, we will provide more explanation about this phenomenon in the experimental part. Thank you for this suggestion.

---

### Official Review · AnonReviewer4 · 2020-10-31
**Official Blind Review #4**

**Rating:** 7
**Confidence:** 3

**Review:**

##########################################################################

Summary:
This paper incorporates AdaBoost into the deep graph neural network architecture, which has the ability to efficiently extract knowledge from high-order neighbors and then integrates knowledge from different hops of neighbors into the network in an Adaboost way. It solves the problem of oversmoothing. Extensive experiments show the effectiveness of the proposed method.

##########################################################################

Pro:
+ The paper is clear and well organized.
+ The introduction of AdaBoost into the deep GNN is novel and interesting.
+ The comparison with several existing methods is well analyzed in terms of both model architectures and computational advantages.
+ Extensive experiments are conducted to demonstrate the consistent state-of-the-art performance of the proposed method.

##########################################################################

Cons:
- Although the same classifier architecture is adopted for $f^{(l)}_\theta$, their parameters are different, which is different from RNN. It is better to avoid this confusion.

- It would be better to include some discussion of the global attention methods (e.g., [Puny et al., 2020] ) and sampling-based methods (e.g., [Zeng et al., 2020]).

References:
Puny et al. From Graph Low-Rank Global Attention to 2-FWL Approximation. ICML Workshop Graph Representation Learning and Beyond, 2020.

Zeng et al. Graph sampling-based inductive learning method. ICLR ’20, 2020.

---

> ### Author Response · Authors · 2020-11-16
> **Response to Reviewer 4**
>
> We thank you for your recommendation for acceptance. Here is our clarification.
>
> 1. RNN-like architecture.
>
> You are right that each classifier shares the same architecture, but their parameters are different. Thus, we claim that AdaGCN is only an RNN-like architecture and we will highlight this difference from RNN in the revised version.
>
> 2. Some discussion.
>
> Thank you for your suggestion. We will include more discussion on the global attention methods and sampling-based methods in the revised version.

---

### Public Comment · ~Meng_Liu3 · 2020-11-15
**Related work**

Great work! The proposed approach is very effective and insightful for leveraging information from different hops of neighbors.

I would like to mention our recent work “Towards Deeper Graph Neural Networks (KDD2020) ( https://arxiv.org/abs/2007.09296 )”, which also investigates deep models to utilize information from large receptive fields. This is highly related to your work.

Thank you.

---

> ### Author Response · Authors · 2020-11-16
> **Thank you!**
>
> Thank you. Your recent work is also insightful and effective, and it is also strongly linked with our work. We will cite your work in our related work part.

---

### Decision · Program_Chairs · 2021-01-07
**Final Decision**

**Decision:**

Accept (Poster)

**Comment:**

Three of the reviewers are very positive about this work, and R3 is slightly concerned about the datasets, writing, and notations etc. The authors responded to these concerns in detail and have agreed to take care of these comments. Thus an accept is recommended based on the understanding that the authors will fulfil their commitments.